# Macrophage Polarization Related to Biomimetic Calcium Phosphate Coatings: A Preliminary Study

**DOI:** 10.3390/ma16010332

**Published:** 2022-12-29

**Authors:** Jiping Chen, Yiwen Zhou, Xingnan Lin, Huang Li

**Affiliations:** 1Department of Stomatology, Nanjing Drum Tower Hospital, The Affiliated Hospital of Nanjing University Medical School, No. 321 Zhongshan Road, Nanjing 210003, China; 2Nanjing Stomatological Hospital, Medical School of Nanjing University, No. 30 Zhongyang Road, Nanjing 210008, China; 3School/Hospital of Stomatology, Zhejiang Chinese Medical University, No. 548 Binwen Road, Hangzhou 310053, China; 4Orthodontic Department, Nanjing Stomatological Hospital, Medical School of Nanjing University, No. 30 Zhongyang Road, Nanjing 210008, China

**Keywords:** biomimetic calcium phosphate coating, macrophage polarization, surface topography, ionic environment

## Abstract

Biomimetic calcium phosphate (BioCaP) coatings were used to deliver bone morphogenetic protein 2 (BMP2), and enhance osteogenesis. However, the mechanism for BioCaP coatings interacting with the immune response during bone regeneration remains unclear. In this preliminary study, the effect of BioCaP coatings on macrophage polarization without (BioCaP group) or with BMP2 (BioCaP+Inc.BMP2 group) was investigated. RAW 264.7 cells were cultured on the rough and platelike surfaces of coatings in BioCaP and BioCaP+Inc.BMP2 groups, while cultured on smooth surfaces in the group without material for 5 days. The BioCaP coatings per se modulated the switch of M1 to M2 phenotype from day 3, which promoted the expressions of Arg1 and CD 206 but reduced the expression of TNF-α compared with No material group. To detect the microenvironmental changes, the concentrations of calcium ion (Ca^2+^) and inorganic phosphate (Pi), pH values, as well as calcium phosphate crystal pattern were examined. The trends of ionic environmental changes were closely related with macrophage phenotype switch. These results suggest that BioCaP coating itself may affect the macrophage polarization through surface topography, surrounding ionic environment and calcium phosphate crystal pattern.

## 1. Introduction

Critical sized bone defects weaken the self-healing capacity of bone tissue, leading to defect nonunion. Autografts and allografts still remain the gold standard treatment methods to clinically repair the critical sized defects. However, such strategies have many shortcomings such as risk of infection, graft rejection, donor site morbidity, and limited availability of suitable grafts, encouraging the rapid development of biomaterial-based bone tissue engineering [1,2,3,4,5]. The osteo-conductivity of implanted bone substitute enables the adhesion and ingrowth of differentiated osteoblasts and guarantees the therapeutic efficacy. Biomimetic calcium phosphate (BioCaP) coatings comprised of octacalcium phosphates (OCP) and the beneath amorphous calcium phosphates (ACP) have been developed and used to endow bone substitute biomaterials with osteo-conductivity and enhance the osteogenesis [6,7]. Large amounts of studies have successfully applied BioCaP coatings on the surfaces of different bone substrates, such as titanium-alloy (Ti6Al4V) plates/discs, coralline hydroxyapatite (CHA) granules, deproteinized bovine bones (DBB), PLGA and collagen. The enhanced osteogenesis was also observed in different studies [8,9,10,11,12,13]. However, the biological mechanism of BioCaP coatings to improve osteogenesis was needed to be further studied.

The immune system was reported to closely interact with bone cells and regulate the osteogenesis and prognosis of the regeneration process [14,15,16] after bone substitutes were implanted. As an essential part of the immune system, macrophages accumulated around the materials and activated as two functional types [17,18,19]. The classically activated macrophages were referred as M1 macrophages [19,20,21] and secreted pro-inflammatory cytokines such as interleukin 1 (IL-1) and tumor necrosis factors α (TNF-α). M1 macrophages also produced reactive oxygen and nitrogen intermediates, including NO and superoxide, that are highly toxic for microorganisms but can also be highly damaging to surrounding tissues and lead to aberrant inflammation. The inflammatory response initiated the process of osteoclast-genesis, making this phase indispensable. The alternatively activated macrophages were referred as M2 macrophages [19,20,21] and exhibited the anti-inflammatory capacity and antagonism effects for M1 macrophage responses, which may be crucial for the activation of the wound healing and the restoration of tissue homeostasis [22]. Thus, the timely and effectively conversion of M1 phenotype into the anti-inflammatory M2 phenotype determined the re-establishment effect of bone regeneration.

Meanwhile, macrophages could fuse to form foreign body giant cells (FGBCs) and were considered the predominant drivers during foreign body responses (FBRs) [17,23]. FBRs were reported as key in determining the outcome after materials implanted, making it crucial to understand the FBRs [24]. Lin et al. applied the BioCaP coatings with the coprecipitation of bone morphogenetic protein 2 (BMP2) (BioCaP-Inc.BMP2) on the surface of CHA granules [10] and observed both enhanced osteogenesis and reduced FBRs around the surface of BioCaP-Inc.BMP2. However, the underlying immunomodulation effects of macrophage polarization on the surface of BioCaP-Inc.BMP2 during the FBRs have not been further investigated yet.

In order to figure out whether it is the nature of the BioCaP coating or the presence of BMP2 that dampens the FBRs and influences the underlying macrophage phenotype shifts, we designed this study to preliminarily investigate the immunomodulation effect of BioCaP coatings with or without BMP2 on the regulation of macrophage polarization. The RAW 264.7 cells were cultured on the surface of BioCaP coatings and the immunomodulation effects were demonstrated via changes of RAW 264.7 cells behaviors.

## 2. Materials and Methods

### 2.1. Fabrication of BioCaP Coatings and the Application of BMP2

Sterilized Ti discs were used as substrates. Initially, Ti discs were immersed in the 5-times-concentrated simulated body fluid (683.8 mM NaCl, 12.5 mM CaCl_2_•2H_2_O, 5.7 mM Na_2_HPO_4_•2H_2_O, 20.9 mM NaHCO_3_, 7.5 mM MgCl_2_•6H_2_O) at 37 °C for 24 h. Thus, a thin (1–3 μm) and dense layer of ACP was formed on the surface of Ti discs. The OCP layer was deposited on the surface of ACP layer by immersing the samples (10 mL per sample) for 48 h at 37 °C in a supersaturated solution of calcium phosphate (4.0 mM CaCl_2_•2H_2_O, 136.8 mM NaCl, 2.3 mM Na_2_HPO_4_•2H_2_O), buffered with 49.9 mM TRIS. BMP2 (Shanghai Rebone Biomaterials Co., Ltd., Shanghai, China) was added in the OCP solution and incorporated into the internal crystalline of BioCaP coating. The whole coating process was performed under sterile conditions and the inorganic agents mentioned above were purchased from Sigma-Aldrich Corporation (St. Louis, Missouri, USA).

### 2.2. Scanning Electron Microscopy (SEM)

The surface structural characterizations were accomplished by SEM (S-3400N, Hitachi, Ibaraki, Japan) after gold-sputtered.

### 2.3. Cell Culture, Stimulation and Treatment

The mouse macrophages (RAW 264.7 cells) cell line was obtained from Dr. Bijun Zhu. RAW 264.7 cells were cultured in the complete culture medium (DMEM, Gibco, New York City, New York, USA) with 10% fetal bovine serum (FBS, Gibco, New York City, New York, USA) and 1% penicillin-streptomycin (Hyclone, Logan, Utah, USA) at 37 °C in a 5% CO_2_ incubator. After seeded on the surface of BioCaP coatings, the RAW 264.7 cells were stimulated with lipopolysaccharide (LPS, Sigma (055: B5), St. Louis, Missouri, USA) and interferon gamma (IFN-γ, T&L Biological Technology, Beijing, China) for 24 h into pro-inflammatory M1 macrophages (as illustrated in Figure 1). At the different time points, cell lysates, the supernatant of the cultured medium, and the soaked BioCaP coatings were harvested.

### 2.4. Related Gene Expressions of M1 and M2 Macrophages

The total RNA was extracted using RNA Extraction Kit (Beyotime, Haimen, China) and transcribed with a cDNA Reverse Transcription kit (Vazyme, Nanjing, China). Quantitative polymerase reaction was performed using SYBR Green Mix (Vazyme, Nanjing, China). The RNA expressions were examined for the following factors: TNF-α (M1 macrophages markers), Arg-1 and CD206 (M2 macrophages markers). GAPDH was used as the internal reference and the primer sequences are listed in Appendix A.

### 2.5. Measurement of Changes in the Ionic Microenvironment in the BioCaP Coating Treated Macrophage Culture Medium

The culture medium of the macrophages seeded on the surface of BioCaP coating was collected at the different time intervals. The concentration of calcium ion (Ca^2+^) and inorganic phosphate (Pi) ions in the medium were quantitatively tested with Calcium Colorimetric Assay Kit (Beyotime, Haimen, China) Micro Blood Phosphorus Concentration Assay Kit (Solarbio, Beijing, China), respectively. The pH value of the collected culture medium was determined with a pH meter (METTLER TOLEDO, Zurich, Switzerland).

### 2.6. Fourier Transform Infra-Red Spectroscopy (FTIR) Analysis of BioCaP Coatings

BioCaP coatings cultured in the medium with RAW 264.7 cells for 5 days were collected and evaluated by attenuated total reflectance (ATR) model in a FTIR instrument (IS5, Thermo Fisher Scientific, Waltham, Massachusetts, USA).

## 3. Results

### 3.1. Responses of RAW 264.7 Cells to the BioCaP Coatings

Figure 2 showed the surface topography of the BioCaP coatings without or with BMP2. As revealed by SEM, the BioCaP coatings consisted of the platelike crystals with sharp edges (Figure 2A), which became smaller, more compact and more bent in the presence of BMP2 (Figure 2B). Then, RAW 264.7 cells were seeded on the rough surface of BioCaP coatings and cultured for 5 days, as illustrated in Figure 1. The phenotype switch was indicated by the related gene markers, TNF-α for M1, and Arg1 and CD206 for M2. The gene expressions of TNF-α, Arg1, and CD206 were displayed in Figure 3. The BioCaP coatings up-regulated the expression of TNF-α at day 1 but down-regulated that after day 3 in both BioCaP group and BioCaP+Inc.BMP2 group compared with the No material group. The expressions of anti-inflammatory genes (Arg1 and CD206) were significantly decreased at day 1 but increased after day 3.

### 3.2. Ionic Microenvironment Changes of Culture Medium in the Presence of BioCaP Coatings

The outer components of BioCaP coatings were OCP, which can incorporate Ca^2+^ and release Pi ion when hydrolyzed in the medium, and could affect the functional behaviors of RAW 264.7 cells due to the ionic concentration changes of macrophage medium. The ion concentrations of Ca^2+^ and Pi, as well as the pH value of the culture medium were detected over 5 days as demonstrated in Figure 4.

The concentration of Ca^2+^ was sharply decreased in BioCaP group and BioCaP+Inc.BMP2 group after 1 day incubation and became slightly higher with time at day 3 and at day 5, but still lower than that in RAW 264.7 cell group. The Pi ion concentration displayed an obvious increase at day 1 but a sharp decrease at day 3 in both BioCaP group and BioCaP+Inc.BMP2 group, which remained lower up to 5 days and gradually back to the level that of the control group. The concentration of Pi ion in RAW 264.7 cell groups was slightly higher than that in culture media over the 5 days. The pH values of BioCaP group and BioCaP+Inc.BMP2 group were increased up to about 8.2 at day 1 and then decreased in the following days. The pH values for both RAW 264.7 cell group and culture medium group were decreased gradually and remained lower than BioCaP group and BioCaP+Inc.BMP2 group over the 5 days.

### 3.3. FTIR Characterizations of BioCaP Coatings

The CaP structural characterizations were resolved by FTIR in Figure 5. The spectra pattern of CaP materials in BioCaP group and BioCaP+Inc.BMP2 group before and after incubation with RAW 264.7 cells for 5 days was detected. Originally, the typical v4 vibration bands of PO_4_ ^3-^ located at 600 cm^−1^ and 560 cm^−1^ in BioCaP group and 600 cm^−1^ and 555 cm^−1^ in BioCaP+Inc.BMP2 group. After 5 days incubation, the bands slightly shifted and the peaks became obscured in both BioCaP group and BioCaP+Inc.BMP2 group, suggesting the crystal phases of the BioCaP coatings were converted due to the presence of RAW 264.7 cells and this change was independent of BMP2.

## 4. Discussion

A vast number of studies have confirmed the superior osteo-conductivity and osteo-inductivity of the BioCaP coatings [8,9,10,25,26]. However, it is still unclear how the BioCaP coatings interact with the host environment and enhance the osteogenesis. Numerous studies have reported an indispensable role of macrophage in the bone regeneration process and the outcome of implanted biomaterials [17,19,27,28], inspiring us to explore the interactions between BioCaP coatings and macrophages. In this preliminary study, BioCaP coatings with or without BMP2 were fabricated and the capacity to immune response was examined with RAW 264.7 macrophages. The expressions of M1 related genes were increased in the early stage (day 1) but decreased later, while the M2 related genes were down-regulated in the early stage (day 1) but up-regulated afterwards in both BioCaP group and BioCaP+Inc.BMP2 group. The ion concentration (Ca^2+^ and Pi ion) and pH value for the culture medium were changed in the presence of BioCaP coatings. The FTIR spectra suggested that crystal phases of the BioCaP coatings could been converted when cultured with macrophages. These results suggest the BioCaP coating itself may affect the cellular responses of RAW 264.7 cells via the changes of surface topography, surrounding ionic environment and calcium phosphate crystal phases, which was independent of the presence of BMP2.

The physical cues of biomaterials have been reported to influence the behaviors of macrophages [17,29]. The surface topography is one of the essential cues that can be sensed by the surrounding macrophages. Biomaterials with rough surfaces enhanced the phagocytic property of different macrophages, such as THP-1 cells [30] and J774A.1 cells [31]. Rough surfaces were also reported to be more capable of activating RAW 264.7 cells towards a M2 phenotype [32]. Moreover, the surfaces with microscale topographies triggered less pro-inflammatory responses in macrophages derived from human THP-1 cells than nano topographies [33]. The BioCaP coatings studied in this present research have been reported as a rough layer comprised of 30~50 µm straight, platelike calcium phosphate crystal [13,34,35]. However, the macrophage inflammatory response in this study was first slightly enhanced by BioCaP coating in the early stage but weakened in the following days, rather than simply remaining the M1 or M2 phenotypes after activation, suggesting the sophisticated response of macrophage to the biomaterial surface topography.

The change of inorganic ion environment could also be one of the physical cues that affected the response of macrophages. The supernatant Ca^2+^ was reported to be sensed by RAW 236.7 cells to enhance the expression of anti-inflammatory genes through the calcium-sensing receptor pathway. In this study, the converted crystal phases of the BioCaP coatings may affect the co-precipitation of the supernatant Ca^2+^ and the concentration of Ca^2+^ was decreased, which explained the M1 polarization in the early stage in presence of the BioCaP coating. Besides, the pH value of culture medium could affect the polarization behaviors of macrophages [36]. The higher pH (>8.2) was prone to induce the polarization of M1 macrophages. The lower pH (<6.6) was reported to induce the polarization of M2 macrophages. In the present study, the pH values for BioCaP group and BioCaP+Inc.BMP2 group were obviously higher (7.7–8.2) at day 1 compared to RAW 264.7 cell group and culture medium group, which may have stimulated the polarization of M1 macrophages.

Although the BioCaP coating was regarded as biocompatible [10], the main component, OCP, was reported to promote the inflammatory response after initial implantation through stimulating the release of proinflammatory factors, such as IL6 [37]. As demonstrated in this study, the expression of M1 phenotype related gene TNF-α was elevated in the early stage (day 1). The M1 macrophage has been reported to secrete angiogenic (VEGF) and proinflammatory cytokines (IL-6) to promote cell migration and bone defect healing process, making this phase indispensable [21,27,37]. Although the M2 macrophages have been traditionally recognized as a “good” phenotype due to the anti-inflammation capacity, the excessive polarization of M2 macrophages or early transition from M1 to M2 phenotype may lead to delayed defect healing, scar formation, nonunion wound and poorly vascularization [27,38]. The BioCaP coatings can trigger a short inflammatory stage (day 1) and promote the shift from M1 to M2 later (day 3 and day 5) with or without BMP2, indicating the BioCaP coating itself may participate the osteo-immunomodulation process to enhance osteogenesis.

In recent decades, the exploration of osteo-immune responses has been an exciting topic, trying to elicit the interactions between bone cells and immune cells. In this preliminary study, the RAW 264.7 cells were cultured on the surface of BioCaP coatings and the immunomodulation effects were demonstrated via changes of related gene expressions, physical environment (ionic concentration and pH value) and calcium phosphate pattern. While the more specific signals, such as the alignment of the cytoskeleton and signaling cascades, remain to be further investigated.

## 5. Conclusions

In this preliminary study, we found BioCaP coatings created positive immunomodulation effects for macrophages to provoke slight inflammatory reactions after initial implantation and then convert to the anti-inflammatory phenotype to elicit further bone regeneration.

BioCaP coating itself may affect the macrophage polarization through the changes of surface topography, surrounding ionic environment and calcium phosphate crystal pattern. The osteogenic environment caused by the application of BMP2 seemed not to influence the macrophage phenotype shifts.

Further studies will be needed to demonstrate the more specific mechanisms triggered by BioCaP coatings to modulate the interaction between immune system and bone regeneration.

## Figures and Tables

**Figure 1 materials-16-00332-f001:**
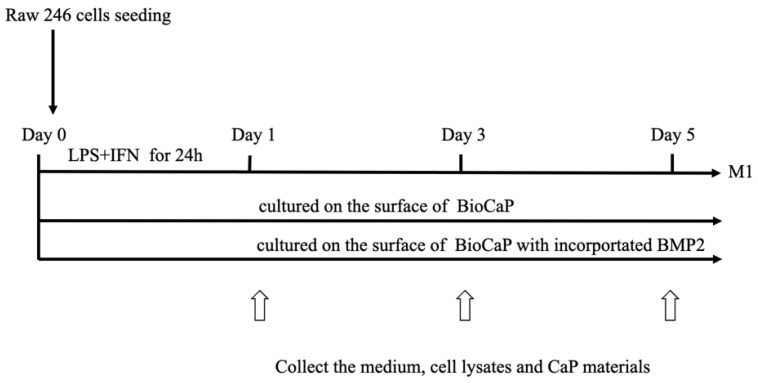
The treatment process of RAW 264.7 cells. Stimulated as M1 macrophages with LPS and IFN-γ for 24 h, the RAW 264.7 cells were cultured on the surface of BioCaP coatings with (M1+BioCaP+Inc.BMP2) or without BMP2 (M1+BioCaP) for 5 days. The cultural medium, cell lysates and BioCaP coatings were collected at 1, 3 and 5 days for further analysis.

**Figure 2 materials-16-00332-f002:**
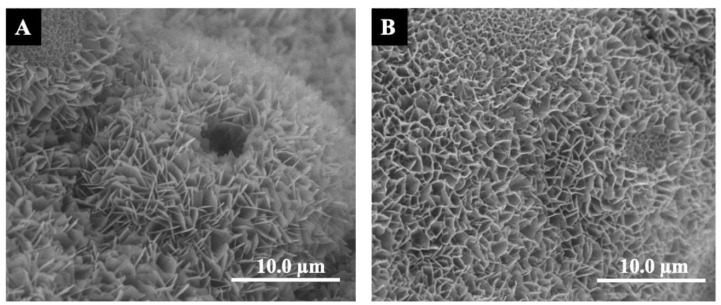
The SEM Scanning electron microscopy (SEM) images of the biomimetic calcium phosphate coatings. (**A**) Surface topography of the BioCaP coatings. (**B**) Surface topography of the BioCaP coatings with incorporated BMP2. The scale bar = 10.0 µm.

**Figure 3 materials-16-00332-f003:**
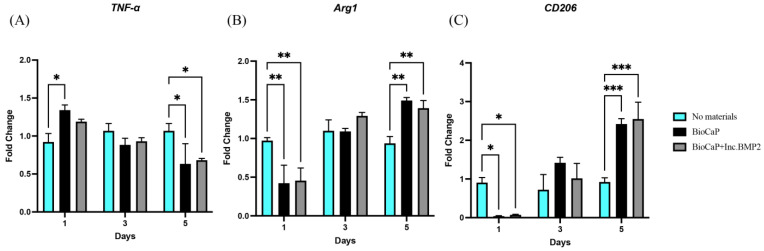
The expressions of M1 (**A**) and M2 (**B**,**C**) related genes in RAW 264.7 cells after co-culturing with BioCaP coatings on the 1st, 3rd and 5th day. * *p* < 0.05, ** *p* < 0.01, *** *p* < 0.001.

**Figure 4 materials-16-00332-f004:**
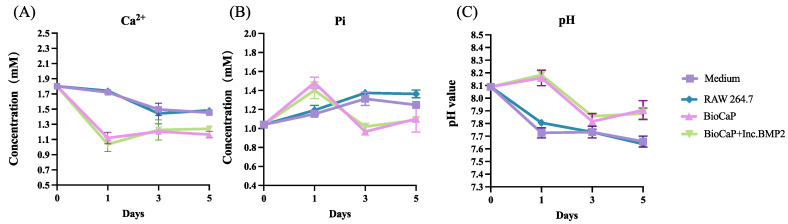
The changes of Ca^2+^ concentration (**A**), Pi concentration (**B**), and pH value (**C**) of culture medium on the 1st, 3rd and 5th day.

**Figure 5 materials-16-00332-f005:**
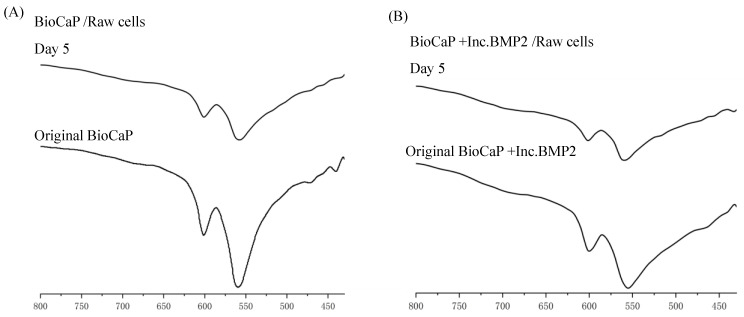
FTIR spectra of BioCaP group (**A**) and BioCaP+Inc.BMP2 group (**B**) before and after incubation with RAW 264.7 cells for 5 days.

## Data Availability

We declare that the data underlying this article was original and will be fully available and obtained from the corresponding author.

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
