# Peer review of "Macrophage Polarization Related to Biomimetic Calcium Phosphate Coatings: A Preliminary Study"

_materials, 2022, doi:10.3390/ma16010332_

Round 1

Reviewer 1 Report

The authors use NO or No trough the manuscript. Please homogenize.

What are the implications of the findings?. The conclusion must include the effect of macrophages' differentiation in bone repairment. Also include the implication of including BMP2 in the formulation.

Author Response

We thank the reviewer for the good comment. We have polished our manuscript carefully and corrected the grammatical, styling, and typos found in our manuscript.

Q1.The authors use NO or No trough the manuscript. Please homogenize.

Reply to Question 1: Thank you very much for the remind and we have homogenized the expressions.

Q2.What are the implications of the findings? The conclusion must include the effect of macrophages' differentiation in bone repairment. Also include the implication of including BMP2 in the formulation.

Reply to Question 2: Thank you very much for your valuable suggestions. The conclusion part has been rewritten to demonstrate our findings in a more specific way.

As regards to the effect of macrophages' differentiation in bone repairment, we found the BioCaP coatings provided a beneficial environment for macrophages to initiate the inflammatory responses and then convert to the anti-inflammatory phenotypes to promote the bone repair.

The previous studies applied the BioCaP coatings with the coprecipitation of bone morphogenetic protein 2 (BMP2) (BioCaP-Inc.BMP2) on the surface of CHA granules and observed both enhanced osteogenesis and reduced FBRs around the surface of BioCaP-Inc.BMP2. However, the underlying immunomodulation effects of macrophage polarization on the surface of BioCaP-Inc.BMP2 during the FBRs have not been further investigated yet. In order to figure out whether it is the nature of the BioCaP coating or the presence of BMP2 that dampens the FBRs and influences the underlying macrophage phenotype shifts, we designed this study and found immunomodulation effect caused by BioCaP coatings seemed to be independent of the presence of BMP2.

According to this comment, the conclusion part has been rewritten as follows:

In this preliminary study, we found BioCaP coatings created positive immunomodulation effects for macrophages to provoke slight inflammatory reactions after initial implantation and then convert to the anti-inflammatory phenotype to elicit further bone regeneration.

BioCaP coating itself may affect the macrophage polarization through the changes of surface topography, surrounding ionic environment and calcium phosphate crystal pattern. The osteogenic environment caused by application of BMP2 seemed not to influence the macrophage phenotype shifts.

Further studies will be needed to demonstrate the more specific mechanisms triggered by BioCaP coatings to modulate the interaction between immune system and bone regeneration.

Changes made: According to this comment, changes were made in page 9, line 330-337.

Reviewer 2 Report

The manuscript titled Macrophage Polarization Related to Biomimetic Calcium Phosphate Coatings: A preliminary study is not very engaging work. The article does not meet the standards of a journal, a minimum of tests have been performed thus I recommend its rejection. It is written in a chaotic manner linguistic and grammatical errors frequently affect its clarity. There are abbreviations in the article that are not explained (e.g. SF). The introduction does not very much explain the sense of the undertaken research and its purpose. The reader only begins to understand, at the end of the article, that the paper has some scientific value. For almost the entire article, the reader reads it with the preconception that the authors pay close attention to the type of coating (with or without BMP2) and try to show the differences between them and their effect on the macrophages (this is how the purpose of the research is presented). However, the reader notes that these differences are not evident in the results. The main result is that the BioCaP coating influences the behaviour of macrophages. It does not matter much whether it is modified with BMP2 or not (Fig. 3, Fig. 4, ).

A few suggestions for changes in order to resubmit or send the article to another journal:

1.       The Control material should be Ti plate.

2.       The Introduction should clearly indicate the meaning and purpose of the research.

3.       The reader is confused about the difference between the study by Lin et al. [18] and this proposed in the manuscript. The novelty must be highlighted.

4.       Certain parts of the results should be in the methods (e.g. Figure 2 and its explanation).

5.       The conclusions are too brief and do not correspond to the stated aim. The objective and conclusions should be rewritten.

Author Response

We thank the reviewer for the good comment. We have polished our manuscript carefully and corrected the grammatical, styling, and typos found in our manuscript.

Q1.The Control material should be Ti plate.

Reply to Question 1: Thank you very much for the remind and we have corrected the accidental writings.

Changes made: According to this comment, changes were made in page 3, line 115.

Q2.The Introduction should clearly indicate the meaning and purpose of the research.

Reply to Question 2: Thank you very much for the valuable suggestions. The Introduction part has been partially rewritten and restructured. The meaning and purpose of the research were corrected as follows:

In order to figure out whether it is the nature of the BioCaP coating or the presence of BMP2 that dampens the FBRs and influences the underlying macrophage phenotype shifts, we designed this study to preliminarily investigate the immunomodulation effect of BioCaP coatings with or without BMP2 on the regulation of macrophage polarization. The RAW 264.7 cells were cultured on the surface of BioCaP coatings and the immunomodulation effects were demonstrated via changes of RAW 264.7 cells behaviors.

Changes made: According to this comment, changes were made in page 3, line 103-105.

Q3.The reader is confused about the difference between the study by Lin et al. [18] and this proposed in the manuscript. The novelty must be highlighted.

Reply to Question 3: Thank you very much for the valuable suggestions. The relationship between the study by Lin et al. and this study has been rewritten more clearly as follows:

Meanwhile, macrophages could fuse to form foreign body giant cells (FGBCs) and were considered the predominant drivers during foreign body responses (FBRs)[17, 23]. FBRs were reported as key in determining the outcome after materials implanted, making it is crucial to understand the FBRs[24]. Lin et al applied the BioCaP coatings with the coprecipitation of bone morphogenetic protein 2 (BMP2) (BioCaP-Inc.BMP2) on the surface of CHA granules[10] and observed both enhanced osteogenesis and reduced FBRs  around the surface of BioCaP-Inc.BMP2. However, the underlying immunomodulation effects of macrophage polarization on the surface of BioCaP-Inc.BMP2 during the FBRs have not been further investigated yet.

Changes made: According to this comment, changes were made in page 3, line 95-102.

Q4.Certain parts of the results should be in the methods (e.g. Figure 2 and its explanation)

Reply to Question 4: Thank you very much for the valuable suggestions. The original Figure 2 has been renamed as Figure 1 and included in the methods part.

Q5.The conclusions are too brief and do not correspond to the stated aim. The objective and conclusions should be rewritten.

Reply to Question 5: Thank you very much for your valuable suggestions. The conclusion part has been rewritten to demonstrate our findings in a more specific way as follows:

In this preliminary study, we found BioCaP coatings created positive immunomodulation effects for macrophages to provoke slight inflammatory reactions after initial implantation and then convert to the anti-inflammatory phenotype to elicit further bone regeneration.

BioCaP coating itself may affect the macrophage polarization through the changes of surface topography, surrounding ionic environment and calcium phosphate crystal pattern. The osteogenic environment caused by application of BMP2 seemed not to influence the macrophage phenotype shifts.

Further studies will be needed to demonstrate the more specific mechanisms triggered by BioCaP coatings to modulate the interaction between immune system and bone regeneration.

Changes made: According to this comment, changes were made in page 9, line 330-337.

Reviewer 3 Report

The manuscript entitled (Macrophage Polarization Related to Biomimetic Calcium Phosphate Coatings: A preliminary study) is interesting. However, some points arise in this article have to be considered before recommendation to be accepted for publication.

1- The language has to be revised as some structural mistakes are present through the whole manuscript.

2- The conclusion section is required to be rewritten in a better and detailed way.

3- The references are not recent enough. Recent relevant references published in 2022 are recommended to be cited in this article.

Author Response

We thank the reviewer for the good comment. We have polished our manuscript carefully and corrected the grammatical, styling, and typos found in our manuscript.

Q1.The language has to be revised as some structural mistakes are present through the whole manuscript.

Reply to Question 1: Thank you very much for your valuable suggestions and the language has been thoroughly revised.

Q2.The conclusion section is required to be rewritten in a better and detailed way.

Reply to Question 2: Thank you very much for your valuable suggestions. The conclusion part has been rewritten to demonstrate our findings in a more specific way as follows:

In this preliminary study, we found BioCaP coatings created positive immunomodulation effects for macrophages to provoke slight inflammatory reactions after initial implantation and then convert to the anti-inflammatory phenotype to elicit further bone regeneration.

BioCaP coating itself may affect the macrophage polarization through the changes of surface topography, surrounding ionic environment and calcium phosphate crystal pattern. The osteogenic environment caused by application of BMP2 seemed not to influence the macrophage phenotype shifts.

Further studies will be needed to demonstrate the more specific mechanisms triggered by BioCaP coatings to modulate the interaction between immune system and bone regeneration.

Changes made: According to this comment, changes were made in page 9, line 330-337.

Q3.The references are not recent enough. Recent relevant references published in 2022 are recommended to be cited in this article.

Reply to Question 3: Thank you very much for your valuable suggestions. Some recent relevant references published in 2022 have been updated in references and the reference numbers are 19, 23, 24, 31 and 32.

Round 2

Reviewer 2 Report

Dear Authors,

Thank you for the answers and corrections. Regarding questions 2, 3 and 4 I am satisfied. Nevertheless, I am still not convinced by the answer to Q1 and indirectly to Q5. 

In Q1 I meant that the control material should be a titanium plate and not a lack of material (no material) as it is written in the abstract or conclusions (Q5) or medium as it was in the studies regarding Ca2+ concentration. Then the importance of surface roughness on macrophage polarisation could even be more pronounced as titanium plate is smoother. 

Author Response

1.The Control material should be Ti plate.

Reply to Question 1: Thank you very much for the valuable suggestion.

Ti was designed as the substrate of the BioCaP coating in this article and the surface of Ti was completely coved with the BioCaP coating. We have corrected some accidental writings in the Method part.

The reasons we did not select Ti plate as the control group were as follows:

  1. According to our previous studies[1-4], the BioCaP coating can only be digested in the presence of osteoclasts. During the experiment period, the macrophages were not induced into osteoclasts, indicating that Ti won’t be exposed and contacted directly with culture media or macrophages.
  2. Meanwhile, some studies[5, 6] reported the smooth surfaces of Ti could induce the macrophages as more M1-like phenotype.

In order to avoid the distractions of Ti plate on the polarization of macrophage and only focus on the effect of BioCaP coatings, we did not use Ti plate as the control group when designed this study.

Your valuable and wonderful suggestions remind us that more nuanced considerations will be needed in the further study so as to obtain a more pronounced result. Thank you very much again.

 Changes made: According to this comment, changes were made in page 2, line 86.

References:

  1. Lin X.;de Groot K.;Wang D.;Hu Q.;Wismeijer D., Liu Y. A review paper on biomimetic calcium phosphate coatings. Open Biomed Eng J 2015, 9:56-64.
  2. Liu T.;Wu G.;Zheng Y.;Wismeijer D.;Everts V., Liu Y. Cell-mediated BMP-2 release from a novel dual-drug delivery system promotes bone formation. Clin Oral Implants Res 2014, 25(12):1412-1421.
  3. Wernike E.;Hofstetter W.;Liu Y.;Wu G.;Sebald H.J.;Wismeijer D.;Hunziker E.B.;Siebenrock K.A., Klenke F.M. Long-term cell-mediated protein release from calcium phosphate ceramics. J Biomed Mater Res A 2010, 92(2):463-474.
  4. Hagi T.T.;Wu G.;Liu Y., Hunziker E.B. Cell-mediated BMP-2 liberation promotes bone formation in a mechanically unstable implant environment. Bone 2010, 46(5):1322-1327.
  5. Yuan Y.;Ren Y.;Dijk M.;Geertsema-Doornbusch G.I.;Atema-Smit J.;Busscher H.J., van der Mei H.C. Phagocytosis and macrophage polarization on bacterially contaminated dental implant materials and effects on tissue integration. Eur Cell Mater 2021, 41:421-430.
  6. Hotchkiss K.M.;Reddy G.B.;Hyzy S.L.;Schwartz Z.;Boyan B.D., Olivares-Navarrete R. Titanium surface characteristics, including topography and wettability, alter macrophage activation. Acta Biomater 2016, 31:425-434.